# Multiresistance states in ferro- and antiferroelectric trilayer boron nitride

Ming Lv[1,5], Jiulong Wang[2,5], Ming Tian[3,5], Neng Wan [3] ✉, Wenyi Tong [2] ✉, Chungang Duan[2,4] & Jiamin Xue [1] ✉

Stacking two atomic layers together can induce interlayer (sliding) ferroelectricity that is absent in their naturally occurring crystal forms. With the flexibility of two-dimensional materials, more layers could be assembled to give rise to even richer polarization states. Here, we show that three-layer boron nitride can host ferro- and antiferroelectric domains in the same sample. When used as a tunneling junction, the polarization of these domains could be switched in a layer-by-layer procedure, producing multiple resistance states. Theoretical investigation reveals an important role played by the interaction between the trilayer boron nitride and graphene substrate. These findings reveal the great potential and unique properties of 2D sliding ferroelectric materials.

Due to interlayer electron orbital distortion, manually stacked two atomic layers of two-dimensional (2D) materials could develop out-of-plane polarization[1-7], whose direction depends on the in-plane atomic registration. Sliding one layer related to the other by a fraction of the unit cell can reverse the polarization, hence giving the name of 'slidetronics' to this system[3,8-10]. This mechanism has been used to explain several experiments that observed ferroelectric behaviors in 2D systems[11,12]. Unlike conventional ferroelectric materials where the atom displacements are parallel to the polarization, here they are perpendicular to each other, providing new routes to manipulate the ferroelectric states. In addition, the layer-stacking method is very flexible as compared with traditional crystal growth. Therefore, it is foreseeable that by stacking more layers different polarization states could be obtained, which have been observed in multilayer 3R $MoS_2$ samples[10,13], manually stacked or epitaxially grown trilayer BN samples[14,15], manually stacked trilayer $WSe_2$ and $MoS_2$[16]. Here, we fabricate trilayer boron nitride (BN) samples and unveil the rich polarization states together with their intriguing switching behaviors in this system with the ferroelectric tunneling process. The ferro- and antiferroelectric domains developed in the trilayer BN samples exhibit layer-by-layer switching behavior, which is rare in other ferro- and antiferroelectric systems. All the possible switching pathways are precisely identified, which reveals the interesting, albeit weak,

interaction between the polarization states located at the different interfaces. The multiresistance states resulted from the layer-by-layer switching process can be utilized in novel devices such as multi-valued memories. With first-principles calculations, we find that not only the internal polarization of the trilayer BN but also the graphite substrate are important in explaining the observed resistance states.

## Results

### Identification of polarization states

The trilayer BN devices are fabricated with the tear-and-stack technique[17] and measured as a ferroelectric tunneling junction with a conductive atomic force microscopy (CAFM), as schematically shown in Fig. 1a. Polarization state dependent tunneling have been used as an important probe to study the ferroelectric or antiferroelectric materials[18,19], and also a potential technology for data storage[20]. In our experiment the tunneling device is fabricated by stacking three BN monolayers (torn apart from the same larger BN monolayer) in a parallel lattice arrangement on a sizeable graphite flake of about 50 nm thick. This graphite flake serves as one of the two electrodes in the tunneling junction, while the CAFM tip functions as the other one. With a sample bias voltage applied to the graphite flake and the CAFM tip grounded, tunneling current can be mapped and has been previously used to probe the ferroelectric

[1]School of Physical Science and Technology, ShanghaiTech University, Shanghai, China. [2]Key Laboratory of Polar Materials and Devices (MOE), Ministry of Education, Department of Electronics, East China Normal University, Shanghai, China. [3]Key Laboratory of MEMS of Ministry of Education, School of Integrated Circuits, Southeast University, Nanjing, China. [4]Collaborative Innovation Center of Extreme Optics, Shanxi University, Taiyuan, Shanxi, China. [5]These authors contributed equally: Ming Lv, Jiulong Wang, Ming Tian. ✉e-mail: wn@seu.edu.cn; wytong@ee.ecnu.edu.cn; xuejm@shanghaitech.edu.cn

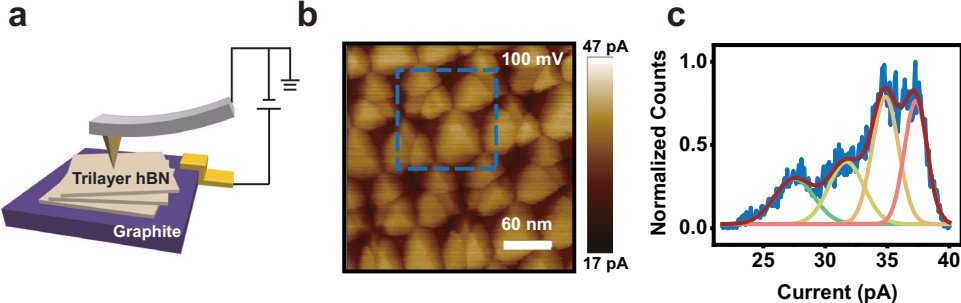

**Fig. 1 | Tunneling current in twisted trilayer BN. a** Schematic diagram of the trilayer BN and tunneling measurement setup. **b** CAFM image with 100 mV sample bias voltage. **c** The histogram of the current values in the area marked by the blue dashed box in b, which is fitted by four Gaussian peaks.

polarization in twisted bilayer BN[21]. Similar current mapping is obtained here on a topographically featureless trilayer BN with 100 mV sample bias (Fig. 1b), revealing an intricate pattern of overlapping triangular domains which have different current values. In the case of twisted BN bilayer, only high and low current domains were observed corresponding to the two out-of-plane polarization states[21,22]. While for the trilayer BN, the histogram of the current values within an area outlined by the blue dashed box in Fig. 1b clearly reveals four peaks as shown in Fig. 1c. This result can be understood as a combination of the polarization states residing between the first and second, or second and third layers of BN, which results in four possible combinations as ↑↑, ↑↓, ↓↑ and ↓↓. Here the ↑ and ↓ designate the out-of-plane polarization and the left (right) ↑ or ↓ is from the interface of the top (bottom) two layers of BN (see Table S1 for a complete notation definition). Recent work on twisted trilayer transitional-metal dichalcogenides has shown that the polarization induced surface potentials are cumulative from that of twisted bilayers[16,23]. This cumulative behavior is also observed in tunneling current measurements. Two overlapping high (low) current domains give the highest (lowest) current, while others give intermediate currents. However, this information alone does not allow us to assign specific polarization combination to each peak shown in Fig. 1c.

To identify the relationship between polarization combinations and resistance states, it is necessary to study the evolution of the domains under an external electric field. Figure 2 depicts two current mappings at 100 mV and 500 mV sample bias voltages. As the voltage is increased the size of small triangles decreases (two of them are outlined by the red dashed lines). Meanwhile, another domain wall (marked by a blue dashed line) arising from a different interface shifts to the left, indicating a decrease in the size of the domain on the left of the blue dashed line. Due to an increase in external electric field directed away from the graphite (sample bias from 100 mV to 500 mV), the dipoles pointing towards the graphite becomes unfavorable and tend to flip upwards. Therefore, the polarization of the small triangular domains (red dashed line) and the large domain on the left of the blue dashed line is pointing down. However, it remains unclear which domain belongs to the top interface and which one belongs to the bottom.

To resolve this issue, we investigate the regions adjacent to the step between the bilayer and trilayer areas (Fig. S1a, b), where it is easy to identify which interface the domains originate from. With the evolution of the domain size as a function of external electric field similar to that in Fig. 2, the polarization combinations are unambiguously determined. Subsequently, a scanning Kelvin probe microscope (SKPM) is used to visualize the surface potential (Fig. S1c, d), which can be ordered from low to high as ↑↑, ↑↓, ↓↑ and ↓↓. This correspondence is then used to identify other trilayer regions far away from the step edges. An example of this process is shown in Fig. S2.

## Resistance evolution in different domains

One area with specific polarization assigned is shown in Fig. 3a. By changing the electric field from Fig. 3a to d, the domain sizes change as expected (with some thermal drift observed in the scans). An interesting feature worth noticing is the changing contrast of the images, which corresponds to different current values in each domain. At low sample bias voltage, the ↑↑ domain has the highest current while ↓↓ has the lowest. Increasing the bias voltage, at some point the contrast between ↑↑ and ↓↑, or ↑↓ and ↓↓ disappears (Fig. 3c and d). This behavior is different from the twisted bilayer BN where the relative current order between the two domains is always the same[21]. To see this evolution more clearly, we take I-V curves at four points represented by the color-coded dots in each domain (Fig. 3a) and present them in Fig. 3e with zoomed-in sections shown in Fig. 3f to h. For the negative bias (Fig. 3f) the current value order from low to high remains as ↓↓, ↑↓, ↓↑ and ↑↑. For the positive bias (Fig. 3g and h), however, there are crossings of currents corresponding to changes of the contrast in Fig. 3a to d. Interestingly, the I-V curves group into two pairs, with ↓↑ and ↑↑ lying above ↓↓ and ↑↓. Similar trend can also be seen in the negative bias range (Fig. 3f) but not as distinct. These behaviors have been repeatedly observed in other samples as shown in Fig. S3. From these observations, it can be inferred that the polarization residing at the lower interface dominates the tunneling process. Hence the interaction between twisted BN and the graphite substrate is important and need to be considered in more depth.

To gain deeper insights into the experimental findings, we incorporate transport calculations using density functional theory with non-equilibrium Green's function (DFT + NEGF) approach. We model the experimental structure as a trilayer BN on graphene. Four stacking types between the trilayer BN and graphene are considered, which are boron-on-carbon (B-C) stacking, nitrogen-on-carbon (N-C) stacking, fully aligned and completely mismatched (Figs. S4a to d). The stacking energy calculations, as depicted in Fig. S4e, reveal that regardless of trilayer BN states the B-C stacking is more energetically favorable than others, in accordance with previous studies[1,24]. As a result, B-C stacking is adopted to calculated the I-V curves. The interlayer distance between graphene and the BN layer above is ~4 Å, greater than the distance between the BN layers (~3.1 Å). Based on these results, the transport models are constructed in Fig.4 with the (111) surfaces of bulk Au as electrodes. Similar to experimental observations, the calculated I-V curves (Fig. 4e) can be separated into two groups. Overall, ↑↑ and ↓↑ have larger current, and ↓↓ and ↑↓ have smaller current. It is clear that the polarization residing between the lower two BN layers determines this overall behavior, which can be seen more clearly in the charge distribution at the BN/graphene interface. With the across-layer effect[24,25] taken into account, charge distribution depends on the interaction between the bottom two layers of BN and graphene substrate. As shown in Fig. S5, differential charge density distributions, in accompany with Hirshfeld charge analysis, confirm that the impact of the graphite substrate on trilayer

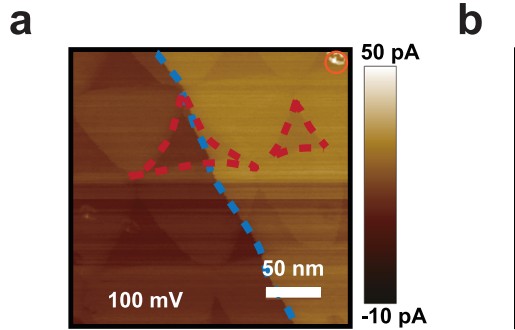
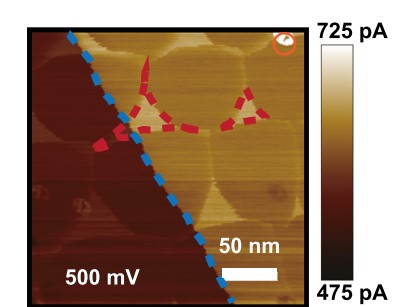

**Fig. 2 | Evolution of the domains under an external electric field. a** and **b** CAFM images with 100 mV and 500 mV sample bias voltages. Two small triangular domains are highlighted by the red dash lines. The blue dashed line marks a domain wall from a different interface. The orange circle points out a surface defect that can be used as a reference of position since the instrument has unavoidable drift during scan.

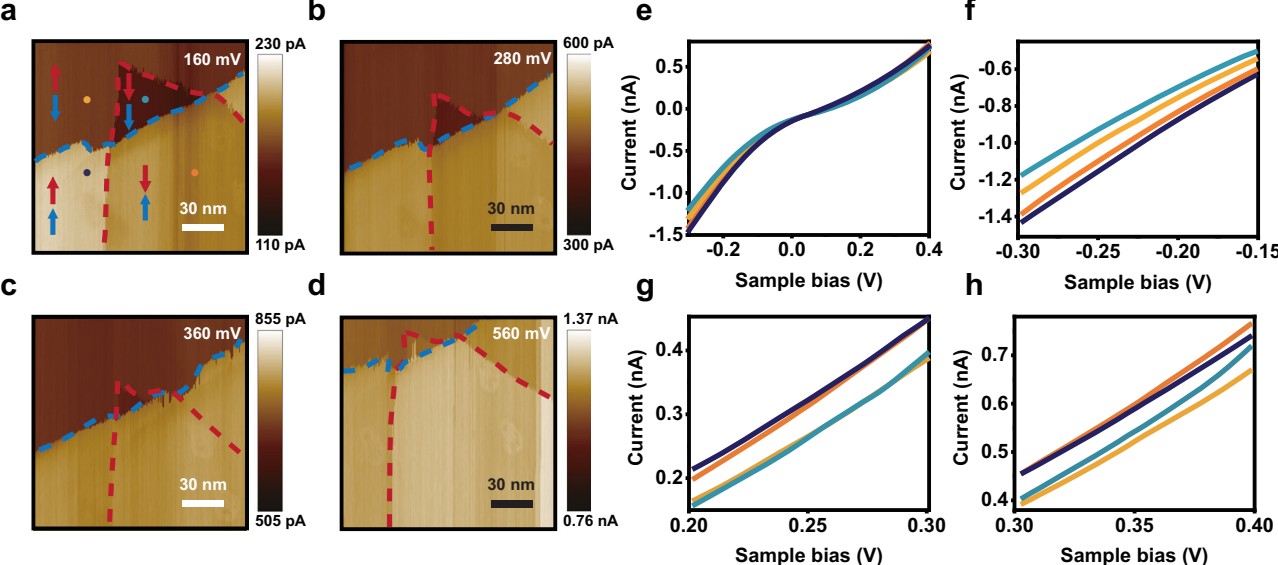

**Fig. 3 | Evolution of current in different domains. a-d** CAFM images with 150 mV, 280 mV, 360 mV and 560 mV sample bias voltages. The domains from the upper (lower) interface are highlighted by red (blue) dash lines, and the red (blue) arrows represent the direction of the upper (lower) polarization. The four color-coded dots indicate the approximate positions where the I–V curves are obtained. **e** Four I–V curves corresponding to the different positions in a. **f–h** Zoomed-in I–V curves of **e**.

BN differs depending on the polarization state of the lower two layers of BN, corroborating experimental observations. Another feature of the calculated result in Fig. 4e is the crossing within each group at voltages similar to those observed in experiment (Fig. 3). It is worth noting that the current in Fig. 4e is about two orders of magnitude smaller than the experimental values in Fig. 3. This is due to the fact that in the calculation, tunneling current was integrated within a unit cell with a radius of several angstroms, while in experiments, the tunneling happened within a radius of several nanometers (estimated from the resolution in current mappings).

Since the BN and graphite are not intentionally aligned in the experiment, other stacking types could also present in the devices. Therefore, we calculated I-V curves for the N-C stacking (Fig. S6a), fully aligned (Fig. S6b) and completely mismatched (Fig. S6c) types. All of them, similar to the most stable B-C stacking configuration, show the same behaviors and agree qualitatively with the experiment.

Further theoretical calculations reveal the important role of graphene. As shown in Fig. S6d, the segregation into two groups and crossing behaviors of the I-V curves all disappear when the graphene layer is removed, which confirms the effect of graphene on the electronic properties. Note that the current in Fig. S6d is significant higher than that in Figs. S6a to c. This is due to the shortened tunneling distance by removal of the graphene layer. It is also worth emphasizing that without graphene, the four polarization states do have different tunneling resistances, but behave differently from the experiment.

We have performed $k_{||}$-resolved transmissions analysis in the 2D Brillouin zone at the Fermi energy to better understand this phenomenon. Without graphene, the transmission patterns are similar for all polarization states and have weak bias voltage dependence (Fig. S7), giving a synchronous evolution of the I-V curves as shown in Fig. S4b. However, upon adding the graphene layer, the transmission patterns in the reciprocal space varies more strongly among the four polarization states (Fig. S8). At 0 bias, the transmission regions primarily locate at the edges of the Brillouin zone. When the applied bias arises from 0.1 V to 0.6 V, the transmission regions gradually shift to the zone center. The evolution rate of transmission greatly depends on the interaction between the graphene and the BN trilayer. We attribute this phenomenon to the discontinuous electronic states between graphene and BN, which leads to the observed segregation and crossing in the I-V curves.

## Polarization switching process

Similar to the twisted bilayer BN, a strong enough electric field can switch the polarization near the domain walls[21]. Figure 5 shows the

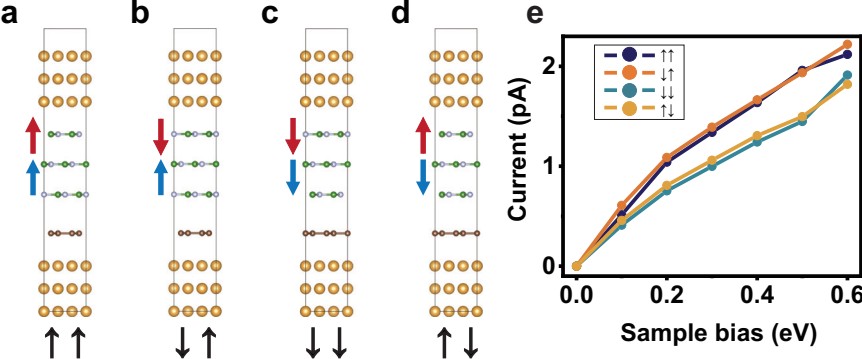

**Fig. 4 | Calculated current evolution in different domains. a–d** Schematic diagrams of the transport model for the ↑↑, ↓↑, ↓↓ and ↑↓ polarization states. The stacking type between the bottom graphene and the BN layer is fixed as the energetically favored B-C configuration. **e** *I–V* curves under different bias. The color coding of different polarization states is consistent with that in Fig. 3.

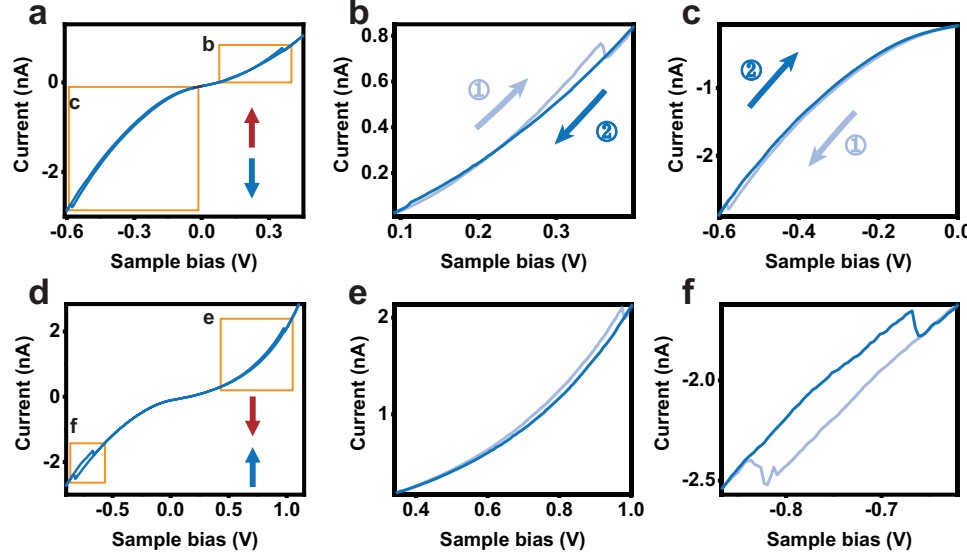

**Fig. 5 | *I–V* curves of the antiferroelectric domains. a**, **d** *I–V* curves measured at the two different antiferroelectric states with the polarization configuration shown in the inset. **b**, **c** Zoomed-in view of the regions marked by the orange boxes in **a**. **e**, **f** Zoomed-in view of the region marked by the orange boxes in **d**. Each *I–V* curve consists of both forward (away from zero bias) and backward (approaching zero bias) scans, which are denoted by the lighter and darker lines, respectively. In **b**, **c** the arrows represent the scanning directions and numbers indicate the scanning sequence. Other following figures are plotted with the same convention.

typical switching process for the antiferroelectric domains. Two hysteresis loops locate at both the positive and negative sample bias sides, corresponding to the switching between the antiferroelectric and ferroelectric states. Zoomed-in sections provide a clearer view of the hysteresis. In conventional antiferroelectric materials, similar two polarization-field loops have been observed[26], sharing a common origin with the current-voltage loops described here.

In contrast, hysteresis loops measured in the ferroelectric domains (Fig. 6) are situated at either the positive (for ↓↓ domains) or negative (for ↑↑ domains) sample bias side. Most of the time, the current jumps twice in either the forward or backward scans, indicating a layer-by-layer switching with an antiferroelectric intermediate state (Fig. 6a, b). Similar switching behavior has recently been observed in 2D antiferromagnets[27,28] but is rare in ferroelectric materials[29]. The possible mechanism could be attributed again to the interaction between the bottom layer and graphite substrate which causes the asymmetry of the coercive fields of the upper and lower polarization. Generally, the bottom polarization has a smaller coercive field so it is flipped first in the forward bias voltage sweep. This can be inferred from Fig. 6 that the first jump is larger than the second one in the forward sweep, and the fact that the switch of the bottom

polarization causes a larger current change as we have learned from Fig. 3. More data of this behavior is shown in Fig. S9.

In some rare cases, the switching paths may differ for the forward and backward voltage sweeps. As depicted in Fig. 6c, in the forward sweep two jumps occur corresponding to the layer-by-layer switching. However, in the backward sweep only one jump is observed corresponding to simultaneous switching of both the upper and lower polarization without an intermediate antiferroelectric state, as schematically shown in the inset. In the 2000 polarization switching curves measured, 16 of them showed this behavior. In another even rarer case (2 out of 2000) shown in Fig. 6d, the backward sweep takes an intermediate state different from that in the forward sweep as depicted in the inset. The rareness of these cases indicates the very weak interaction between the two polarization states located in the two interfaces. As a result, most of the time they can be manipulated by the external electric field independently.

## Discussion
In summary, we have fabricated trilayer twisted BN devices which host both ferro- and antiferroelectric domains in the same sample, indicating the flexibility of sliding ferroelectric systems. Tunneling

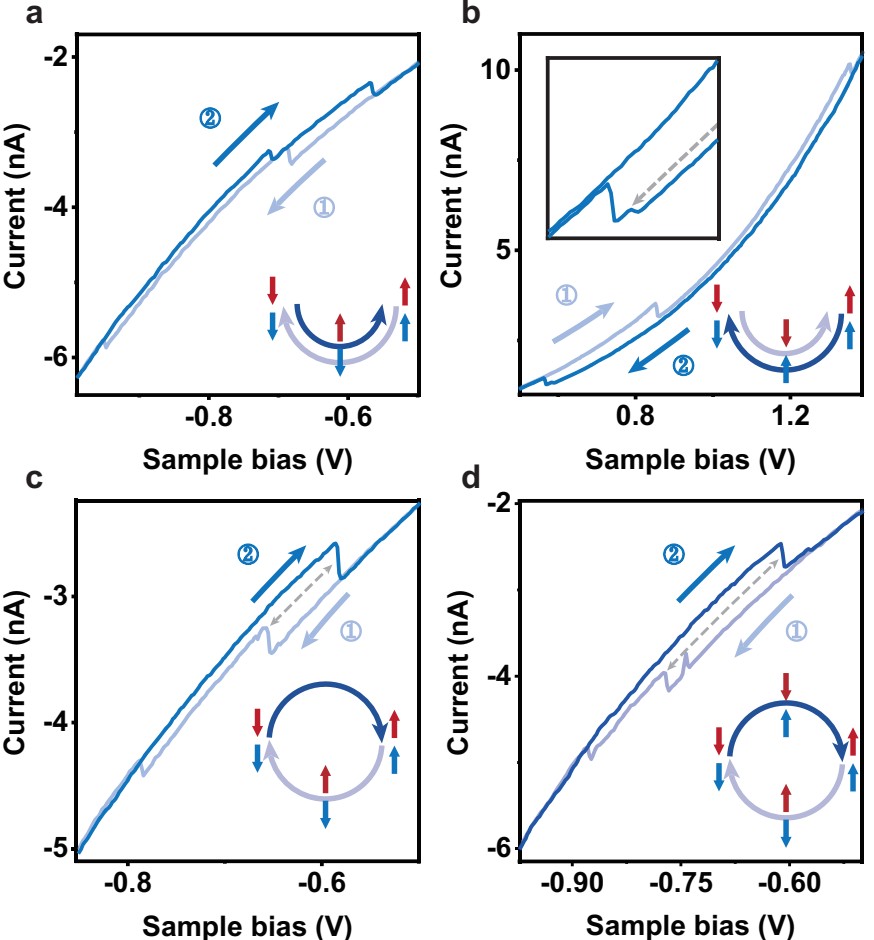

**Fig. 6 | *I–V* curves of the ferroelectricity domains. a**, **b** Most frequently observed *I–V* curves showing layer-by-layer switching. **c**, **d** Cases where the forward and backward sweeps have different intermediate states. Insets: Schematics of the switching paths. The gray dashed double arrows highlight the intermediate states in the forward and backward scans.

measurement revealed multiple resistance states that could be switched to each other with an external electric field. Experiments and theoretical calculations have shown that the rich resistance states originate not only from the polarization within the BN layers, but also from the interaction between BN and the graphite substrate. This work paved a way for future studies in which sliding ferro- and antiferro-electric devices could be fabricated to realize functions such as multi state memories[30] and so on.

## Methods
### Device fabrication
Boron nitride single crystals were grown by an ambient pressure high temperature process[31]. Graphite and BN flakes with appropriate thickness were mechanically exfoliated onto $SiO_2$/Si substrates, and then identified using optical microscopy and AFM. A poly(dimethylsiloxane) (PDMS) stamp covered with polycarbonate (PC) was used to pick up the graphite flakes at 100 °C. Then the graphite could be used to tear and stack the BN monolayers. The desired twist angle (≈0.3°) between the picked up and the remaining BN monolayer can be obtained through a micro stepper motor. With similar tear-twist-stack process, a trilayer BN could be obtained. Due to the wrinkles formed in the stacking process, the final twist angles between the first and second (second and third) layers of BN varied among different areas in the samples, and were determined from the moiré pattern period to be 0.03–0.24°. Finally, the PC film with the flakes was carefully removed from the PDMS stamp and

placed on a $SiO_2$/Si substrate, and then heated at 150 °C to remove the bubbles between the PC film and $SiO_2$/Si substrate. We designed the shape of the electrode with a shadow mask and used electron-beam deposition (Angstrom Engineering deposition system) to deposit metal electrodes (10 nm Ti and 20 nm Au) in vacuum conditions ($<5 \times 10^{-7}$ Torr) at a rate of $0.2 \, \text{Å s}^{-1}$.

### CAFM and FM-KPFM
The CAFM and KPFM images were acquired using Asylum Research Cypher S AFM and Bruker Dimension Fastscan/Icon AFM, respectively. The CAFM tips were MikroMasch NSC15/ Al with a resonance frequency of 325 kHz and spring constant of $40 \, \text{N m}^{-1}$, and the FM-KPFM tips were OPUS 240AC-PP with a resonance frequency of 70 kHz and spring constant of $2 \, \text{N m}^{-1}$. These tips were deposited with 20 nm Ti and 30 nm Au to make them conductive.

### Computational methods
The DFT calculations were carried out with the Vienna Ab initio Simulation Package to optimize the geometry and electronic structure of the models. The projector-augmented wave method was used with a plane wave basis set. The Perdew-Burke-Ernzerhof generalized gra-dient approximation for the exchange correlation functional was adopted. To sample the Brillouin zone, a Γ-centered $8 \times 8 \times 1$ k-points grid was chosen, while a kinetic-energy cutoff of 500 eV was applied for the plane wave expansion. All the structures were optimized until

the Hellmann−Feynman forces were below 1 meV/Å, and the convergence threshold for the electronic energy was $10^{-6}$ eV.

To investigate the device properties, we utilized the DFT + NEGF approach, as implemented in the OPENMX software package. The electron temperature was set at 300 K according to the experimental condition, and $5 \times 5 \times 1$ k mesh was used for the self-consistent calculations to eliminate the mismatch of Fermi level between electrodes and the central region. The conductance was obtained by the Landauer−Buttiker formula at equilibrium:

$$G_0 = \frac{e^2}{h} \sum_{\mathbf{k}} T_\sigma(\mathbf{k}, E_{\mathrm{F}}), \tag{1}$$

and the current is defined as:

$$I = \frac{e^2}{h} \int T_\sigma(E)[f(E - \mu_l) - f(E - \mu_r)]dE \tag{2}$$

The current is determined through the integral at a nonequilibrium state. In this formula, $T_\sigma(\mathbf{k}_\|, E)$ stands for the transmission coefficient with spin $\sigma$, lateral Bloch wave vector $\mathbf{k}_\|$ and energy $E$. $e$ and $h$ are for the electron charge and Planck constant, respectively. In addition, $f$ denotes the Fermi distribution function, while $\mu_l$ and $\mu_r$ indicate the chemical potentials of the left and right electrodes, respectively.

## Data availability
Data presented in this study is available from the authors upon reasonable request.

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

## Acknowledgements

This work was supported by NSFC 12074256 and 12374189 (J.X.), NSFC 12134003 (C.D.), NSFC 12304218 and Shanghai Pujiang Program 23PJ1402200 (W.T.), and NSFC 11674053 (N.W. and M.T.). We thank the Soft Nano Fabrication Center and the Analysis & Characterization Center at ShanghaiTech University.

## Author contributions

M.L. fabricated and measured the trilayer BN samples. J.W., W.T., and C.D. performed theoretical calculations. M.T. and N.W. grew the BN crystals. J.X. conceived the project. All authors contributed to the paper writing.

## Competing interests
The authors declare no competing interests.
