## [Peer Review File · Nature Communications]

Multiresistance States in Ferro- and Antiferroelectric Trilayer Boron NitrideEditorial Note: Figure R1 of this Peer Review File has been redacted as indicated to remove third-party material where no permission to publish could be obtained. Figure R10 of this Peer Review File has been redacted as indicated to maintain the confidentiality of unpublished data.

REVIEWER COMMENTS

Reviewer #1 (Remarks to the Author):

This manuscript reports that a trilayer boron nitride (BN) is able to host ferro- and antiferroelectric domains. The polarization of these domains can be switched in a layer-by-layer procedure, thereby achieving multiresistance states in ferro- and antiferroelectric trilayer BN. This combination of experimental and theoretical approaches provides some valuable insights. This work may be helpful to develop an effective approach for building multiresistance states system in the very interesting 2D samples. Therefore, I would like to recommend it for publication after the authors address the following issues.

1. In bilayer or trilayer BN, how much twist is experimentally required to induce out-of-plane polarization?
2. In Fig. 4, did you consider all the possible interface stacking order between graphene and BN? will the different stacking order have any effect?
3. By comparing Fig. 4 and Fig. 5, it appears that the experimentally observed current and the theoretical predictions exhibit significant differences under the same voltage. What might be the cause of this disparity?
4. In the Computational methods section, the manuscript mentions “The electron temperature was set at 300 K”. Could you please clarify why 300K was set, and whether different temperatures would impact the transport results?

Reviewer #2 (Remarks to the Author):

Interfacial ferroelectrics, whose polarization can be switched by interlayer sliding motion, has been studied intensively in the past few years. While such ferroelectricity has been demonstrated in parallel-stacked bilayer BN and rhombohedral-stacked transition metal dichalcogenides, the antiferroelectricity in the multilayer systems has not been systematically studied as far as I am aware of. The authors made parallel-stacked trilayer BN and studied the switching behavior of its tunneling junction (tip-trilayer BN-graphite) in cAFM. They demonstrate that the BN-graphite substrate interface dominates the signal contrast, and show different switching behaviors between ferroelectric and antiferroelectric states. This study fulfills an important complementary part of the interfacial ferroelectricity, and I think it has potential for publication in Nature Communications. I would like the authors to clarify the following questions before I can recommend the publication.

1. There are too few references covered in the paper, given that there have been many studies in recent years. For example, [Nano Lett. 2023, 23, 15, 7228–7235] shows how the antiferroelectric states distribute during exfoliation. The authors should also include references about ferroelectric tunneling junctions, which is the key structure in this manuscript. For the interfacial ferroelectricity, there are earlier papers on Td WTe₂. These are just a few examples. Please do a more thorough literature review.
2. From line 88 to line 95, the authors mention how they find correspondence between contrast and layer order. However, it is not as clear from the supplementary figures. It is

better to draw some guidance to the eye and explain more in detail to help readers understand their comment.

3. The authors claim that B-C stacking is more favorable when BN and graphite are stacked. However, I think it only makes a difference when BN and graphite are aligned. Is it the case in the experiment? If not, I expect the lattice relaxation is not as significant and BN cannot have all the sites stacked as B-C on graphite.

4. The authors performed theoretical calculations to explain the reason why graphite-BN interface dominates the signal contrast. Although the theoretical calculations seem to agree with the experiment, I wonder if the authors can also give some intuitive arguments in the manuscript. Is it mainly because the tip can be considered as a perfect metal, while the band structure in graphite affects the tunneling probability depending on the polarization of nearest BN bilayer?

Reviewer #3 (Remarks to the Author):

The authors show that different polarisation domains can exist in three rotated hBN layers. The work is interesting, although the results could be better explained.

I have some questions or comments for the authors.

1. The stacking between the three hBN layers is not completely clear; apparently, between the 1-2 is 0.3° , but the angle between the 2-3 is not indicated.

2. The notation of the regions with up/down arrows can be confusing as the right arrow would indicate the interface; the notation leads intuitively to think of areas with aligned or unaligned polarisations. For instance, I would not know what symbology to use when polarisations are opposite at different interfaces, creating a net zero polarization.

3. Labeling the stacks in different regions would be suitable according to the relative stacking between the hBN layers. For example, the ABA would not produce any net polarisation by symmetry, but an ABC or ACB would do.

4. The above can also be applied to DFT calculations (Fig. 4).

5. In the calculations of Fig S4, are the same lattice vectors used for hBN and graphene?

The most probable scenario is that the two crystals are not aligned.

6. Can you explain why, in the hBN bilayer region, there is no dependence of the tunneling current on the lower graphene layer?

7. There is an error in the labeling of Fig S6.

Response letter to reviewers

(Article No. NCOMMS-23-44160)

We are deeply grateful to the reviewers for their efforts in reviewing our manuscript and providing the insightful comments. We are encouraged by the positive remarks from all three reviewers and have improved the manuscript according to their very helpful suggestions.

Reviewer #1 (Remarks to the Author):

This manuscript reports that a trilayer boron nitride (BN) is able to host ferro- and antiferroelectric domains. The polarization of these domains can be switched in a layer-by-layer procedure, thereby achieving multiresistance states in ferro- and antiferroelectric trilayer BN. This combination of experimental and theoretical approaches provides some valuable insights. This work may be helpful to develop an effective approach for building multiresistance states system in the very interesting 2D samples. Therefore, I would like to recommend it for publication after the authors address the following issues.

Response: We greatly appreciate the reviewer's support of our work! We thank him/her for considering our work as valuable and helpful to the research community.

Comment 1-1. In bilayer or trilayer BN, how much twist is experimentally required to induce out-of-plane polarization?

Response 1-1: In the as grown bulk BN, the neighbouring layers are rotated by 180 degrees related to each other (Fig. R1 A). With symmetry restrictions, polarization is not allowed in the out-of-plane direction in this case. For the manually stacked bilayer and trilayer BN samples (Figs. R1B and C), however, ideally zero degree twist is needed to realize the AB or BA stacking which can induce the out-of-plane polarization. Fortunately, for small twist angles (less than ~ 2 degrees), lattice reconstruction happens which maximizes the AB and BA stacking regions [Nat. Nanotechnol. 15, 592 (2020)] and accommodates the twist in the domain walls (as schematically shown in Fig. R1D and experimentally seen in Figs. R1E and F). In our work, we chose the target twist angle to be 0.3° . Due to the wrinkles formed in the stacking process, the final twist angles

varied between different areas in the samples, and were determined from the Moire pattern period to be $0.03^\circ \sim 0.24^\circ$.

[Redacted]

Fig. R1. (A) Illustration of the atomic arrangement for AA' stacking, the bulk form of hBN. Nitrogen and boron atoms are shown in silver and green, respectively. (B and C) Illustration of the atomic arrangement for AB and BA stacking. The vertical alignment of nitrogen and boron atoms distorts the $2p_z$ orbital of nitrogen (light blue), creating an out-of-plane electric dipole. (D) Illustration of a small-angle twisted bilayer BN after the atomic reconstruction. (E and F) Vertical PFM phase and amplitude images of twisted bilayer BN. Scale bars are 100 nm. [Adapted from Science 372, 1458-1462 (2021)]

Changes made: We have added the following sentences to the Methods section.

“Due to the wrinkles formed in the stacking process, the final twist angles varied between different areas in the samples, and were determined from the Moire pattern period to be $0.03^\circ \sim 0.24^\circ$.”

Comment 1-2. In Fig. 4, did you consider all the possible interface stacking order between graphene and BN? will the different stacking order have any effect?

Response 1-2: We thank the reviewer for bringing up this very important point. To address this comment and related ones from other reviewers, we calculated I - V curves for other three typical stacking orders, i.e. N-C stacking, fully aligned, and completely mismatched. The corresponding lattice configurations are shown below in Fig. R2.

Fig. R2. Top and side views of the atomic arrangements between the bottom BN and the graphene layers, showing the B-C stacking, N-C stacking, fully aligned and completely mismatched configurations.

The calculated I - V curves are plotted below as Fig. R3. It can be seen from Figs. R3a to c that all these stacking orders give similar I - V behaviors although details vary between them. First, the four curves form two groups, and the polarization from the lower two BN layers plays a dominant role in affecting the current. Second, within each group the I - V curves can cross, so that the contrast in current mappings can change as the sample bias voltage is swept. These results agree qualitatively with experimental observations. As a comparison, Fig. R3d plots the I - V curves without graphene, which do not show the two properties stated above. Note that the current in Fig. R3d is significantly higher than that in Figs. R3a to c. This is due to the shortened tunneling distance by removal of the graphene layer.

In all these stacking types calculated, the $\sim 1.8\%$ lattice constant mismatch between graphene and BN was not considered following the common treatment in other theoretical works [e.g. ACS Nano, 11, 6382 – 6388 (2017); Adv. Funct. Mater. 33, 2301105 (2023)]. Taking this small mismatch into account would result in large supercells, which would be very challenging for the first-principles calculations.

Fig. R3. Calculated I - V curves of different stacking configurations. a I - V curves of N-C stacking. **b** I - V curves of B-C stacking. **b** I - V curves of the completely mismatched configuration. **d** I - V curves without the graphene layer.

Changes made: In the second paragraph of the section “**Resistance evolution in different domains**”, the sentence “The stacking energy calculations, as depicted in Fig. S4a, reveal that regardless of trilayer BN states, the boron-on-carbon (B-C) stacking configuration between the bottom graphene and BN is more energetically favorable than the nitrogen-on-carbon (N-C) case, in accordance with a previous study.” has been changed to “**Four stacking types between the trilayer BN and graphene are considered,**

which are boron-on-carbon (B-C) stacking, nitrogen-on-carbon (N-C) stacking, fully aligned and completely mismatched (Figs. S4a to d). The stacking energy calculations, as depicted in Fig. S4e, reveal that regardless of trilayer BN states the B-C stacking is more energetically favorable than others, in accordance with previous studies [ACS Nano 11, 6382-6388 (2017), Adv. Funct. Mater. 33, 2301105 (2023)]. As a result, B-C stacking is adopted to calculate the $I-V$ curves. ”.

The new Fig. S4 is displayed below for your convenience.

Fig. S4 Different stacking configurations and their corresponding energies. a-d Top and side views of the four stacking types at the BN and graphite interface considered in this work, which are B-C stacking, N-C stacking, fully aligned (FA) and competently mismatched (CM), respectively. **e** Calculated stacking energies of these four types for the four different polarization combinations. B-C stacking always has the lowest energy.

The following discussion about other stacking types is added after this paragraph.

“Since the BN and graphite are not intentionally aligned in the experiment, other stacking types could also present in the devices. Therefore, we calculated $I-V$ curves for the N-C stacking (Fig. S6a), fully aligned (Fig. S6b) and completely mismatched (Fig.S6c) types. All of them, similar to the most stable B-C stacking configuration, show the same behaviors and agree qualitatively with the experiment.”

The new Fig. S6 is displayed below for your convenience.

Fig. S6. Calculated I - V curves of different stacking configurations. a I - V curves of N-C stacking. **b** I - V curves of B-C stacking. **b** I - V curves of the completely mismatched configuration. **d** I - V curves without the graphene layer.

Comment 1-3. By comparing Fig. 4 and Fig. 5, it appears that the experimentally observed current and the theoretical predictions exhibit significant differences under the same voltage. What might be the cause of this disparity?

Response 1-3: We thank the reviewer for this excellent question. The current is proportional to the tunneling area. In the calculation, tunneling current was integrated within a unit cell with a radius of several angstroms. In experiments, however, the

tunneling happened within a radius of several nanometers (estimated from the resolution in current mappings). We believe that this is the major factor contributing to the disparity. Other minor factors include that the interlayer distance used in calculation was not exactly the same as that in the experiments, multilayer graphite used in experiments was replaced by graphene, and the influence of a tip was represented by Au leads in our calculations etc, which may influence the tunneling current density.

Changes made: We have added the following discussion to page 5 of the manuscript.

“It is worth noting that the current in Fig. 4e is about two orders of magnitude smaller than the experimental values in Fig. 3. This is due to the fact that in the calculation, tunneling current was integrated within a unit cell with a radius of several angstroms, while in experiments, the tunneling happened within a radius of several nanometers (estimated from the resolution in current mappings).”

Comment 1-4. In the Computational methods section, the manuscript mentions “ The electron temperature was set at 300 K ” . Could you please clarify why 300 K was set, and whether different temperatures would impact the transport results?

Response 1-4: We thank the reviewer for this question which helps us to improve the clarity of our manuscript.

Our experiments were carried out at room temperature of 300 K. As a result, calculations were set at the same condition. To examine whether temperature would impact the transport, we calculated the I - V curves for the $\uparrow\uparrow$ polarization state with B-C stacking at two other temperatures (290 K and 310 K). As shown below, the I - V curves at different temperatures almost coincide, reflecting the robustness of the calculated results. Temperatures far away from ~ 300 K are interesting both experimentally and theoretically, but they are beyond the scope of the current work and hence are not considered.

Fig. R4. I - V curves for the $\uparrow\uparrow$ polarization state at 290 K, 300 K, and 310 K, respectively. The stacking type between the bottom BN and graphene is fixed as the energetically favored B-C configuration.

Changes made: We have added the following sentence to the Computational methods section.

“The electron temperature was set at 300 K according to the experimental condition”.

Reviewer #2 (Remarks to the Author):

Interfacial ferroelectrics, whose polarization can be switched by interlayer sliding motion, has been studied intensively in the past few years. While such ferroelectricity has been demonstrated in parallel-stacked bilayer BN and rhombohedral-stacked transition metal dichalcogenides, the antiferroelectricity in the multilayer systems has not been systematically studied as far as I am aware of. The authors made parallel-stacked trilayer

BN and studied the switching behavior of its tunneling junction (tip-trilayer BN-graphite) in cAFM. They demonstrate that the BN-graphite substrate interface dominates the signal contrast, and show different switching behaviors between ferroelectric and antiferroelectric states. This study fulfills an important complementary part of the interfacial ferroelectricity, and I think it has potential for publication in Nature Communications. I would like the authors to clarify the following questions before I can recommend the publication.

Response: We express our sincere appreciation to the reviewer for his/her positive feedback on our manuscript.

Comment 2-1. There are too few references covered in the paper, given that there have been many studies in recent years. For example, [Nano Lett. 2023, 23, 15, 7228 – 7235] shows how the antiferroelectric states distribute during exfoliation. The authors should also include references about ferroelectric tunneling junctions, which is the key structure in this manuscript. For the interfacial ferroelectricity, there are earlier papers on Td WTe₂. These are just a few examples. Please do a more thorough literature review.

Response 2-1: We are extremely grateful to the reviewer for the careful reading of our manuscript. We have incorporated more references as suggested.

Changes made: The following references are added and discussed. They are numbered as they appear in the manuscript.

I. Antiferroelectric state in 3 R MoS₂

13. Shear Strain-Induced Two-Dimensional Slip Avalanches in Rhombohedral MoS₂. Nano Lett. 23, 7228-7235 (2023).

II. Interfacial ferroelectricity

5. Interfacial ferroelectricity in rhombohedral-stacked bilayer transition metal dichalcogenides. Nat. Nanotechnol. 17, 367-371 (2022).

6. Interfacial ferroelectricity in marginally twisted 2D semiconductors. Nat. Nanotechnol. 17, 390-395 (2022).

7. Ferroelectricity in untwisted heterobilayers of transition metal dichalcogenides. Science 376, 973-978 (2022).

11. Ferroelectric switching of a two-dimensional metal.

Nature 560, 336-339 (2018).

12. Unconventional ferroelectricity in moiré heterostructures.

Nature 588, 71-76 (2020).

III. Ferroelectric and antiferroelectric tunneling

15. Tunneling Across a Ferroelectric.

Science 313, 181-183 (2006).

16. Two-Dimensional Antiferroelectric Tunnel Junction.

Phys. Rev. Lett. 126, 057601 (2021).

17. Ferroelectric tunnel junctions for information storage and processing.

Nat. Commun. 5, 4289 (2014).

IV. Across-layer ferroelectricity

22. Across-Layer Sliding Ferroelectricity in 2D Heterolayers.

Adv. Funct. Mater. 33, 2301105 (2023).

23. Atypical Sliding and Moiré Ferroelectricity in Pure Multilayer Graphene.

Phys. Rev. Lett. 131, 096801 (2023).

Comment 2-2. From line 88 to line 95, the authors mention how they find correspondence between contrast and layer order. However, it is not as clear from the supplementary figures. It is better to draw some guidance to the eye and explain more in detail to help readers understand their comment.

Response 2-2: We thank the reviewer for pointing out this confusing part. We have redrawn the figures and added more explanations in the supporting information, which are copied below for your convenience.

Changes made: Figure S1 and S2 are redrawn and the added explanations are shown in red.

Fig. S1 CAFM mapping near a step between the bilayer-trilayer BN. **a** and **b** CAFM images around the step at 100 mV sample bias voltage. These two images are from the same scan but displayed with different color scales, since the trilayer and bilayer areas have an order of magnitude difference in current. **c** Corresponding KPFM image around the step with a larger scanning area. **The white dashed box corresponds to the area shown in a and b.** **d** Profiles of the potential along the green arrow in **c**.

We use a step edge between the bilayer and trilayer BN to identify the origin of the polarization in different domains. As shown in Figs. S1a and b, the bottom (top) part is the trilayer (bilayer) area. It is evident that the smaller triangular domains come from the bottom interface while the larger ones come from the top. Then the sample bias is swept. The expansion or shrinkage of the triangular domains in response to the sample bias can unambiguously tell the polarization of each domain, e.g. if a domain shrinks when the sample bias is increased, then its polarization is pointing to sample substrate (for more details, see *Adv. Mater.* **34**, 2203990 (2022)). All the polarization states can be fixed this way and are labeled in Figs. S1a and b.

Then SKPM can be used to map out the surface potential. Fig. S1c shows such a scan, with the white dashed box corresponding to the area in Figs. S1a and b. It is a zoomed-out scan compared with Figs. S1a and b to cover a larger area due to the lower resolution of SKPM. A line cut along the green arrow is shown in Fig. S1d with the polarization combinations assigned. It can be seen that the $\uparrow\uparrow$ has the lowest potential while $\downarrow\downarrow$ the highest. The second largest potential comes from the $\downarrow\uparrow$ domain and its value is quite close to that of the $\downarrow\downarrow$ domain. This is understandable since the flip of the bottom polarization that is farther away from the SKPM tip produces a smaller change in potential. This correspondence between the surface potential and polarization can be used to identify other domains. We note that the resistance order of the polarization states at low sample bias voltage can also be used as shown in the discussion about Fig. 3 in the main text. These two methods corroborate each other to confirm the assignment of polarization state to each domain.

Fig. S2 An example of using the SKPM to identify polarization states. **a** and **b** CAFM images at 100 mV and 500 mV sample bias. The ferroelectric domains can be easily identified from the change of their shapes, and are marked out in **b**. **c** KPFM image around the same area with a zoomed-out scan. The numbers are averaged surface

potentials at each domain. The white dashed box corresponds to the area in **a** and **b**. **d** A CAFM image with all the polarization states identified. The orange circles in **a**, **b** and **d** highlight a defect to show the thermal drift during scan.

Fig. S2 shows an example of polarization identification with the help of KPFM. Figs. S2a and b are CAFM images at 100 mV and 500 mV, respectively. **The ferroelectric domains are easily identified from their size change, e.g. the triangular domain on the top left side has its polarization pointing down since it shrinks. On the other hand, the antiferroelectric domains need more information to determine their specific states.** A KPFM scan around this area maps out the surface potentials **as shown in Fig. S2c, with the white dashed box corresponds to the area in Figs. S2a and b. Note that in the KPFM scan no sample bias was applied so the domain sizes and shapes are different from those in Figs. S2a and b.** According to the information from Fig. S1, all the polarization states can now be fixed as shown in Fig. S2d.

Comment 2-3. The authors claim that B-C stacking is more favorable when BN and graphite are stacked. However, I think it only makes a difference when BN and graphite are aligned. Is it the case in the experiment? If not, I expect the lattice relaxation is not as significant and BN cannot have all the sites stacked as B-C on graphite.

Response 2-3: We thank the reviewer for bringing up this very important point. As previously revealed [Nat. Phys. 10, 451-456 (2014)], when BN and graphite are aligned or nearly aligned within $\sim 1^\circ$, the two lattices become commensurate. In our experiment, the BN and graphite were not intentionally aligned, as a result BN may not necessarily have all the sites stacked as B-C on graphite, exactly as the reviewer pointed out. To explore the effect of other stacking types on the current transport, we have calculated three more stacking types, i.e. N-C stacking, fully aligned, and completely mismatched. The corresponding lattice configurations are shown below in Fig. R5.

Fig. R5. Top and side views of the atomic arrangements between the bottom BN and the graphene layers, showing the B-C stacking, N-C stacking, fully aligned and completely mismatched configurations.

The calculated I - V curves are plotted below as Fig. R6. It can be seen from Figs. R6a to c that all these stacking orders give similar I - V behaviors although details vary between them. First, the four curves form two groups, and the polarization from the lower two BN layers plays a dominant role in affecting the current. Second, within each group the I - V curves can cross, so that the contrast in current mappings can change as the sample bias voltage is swept. These results agree qualitatively with experimental observations. As a comparison, Fig. R6d plots the I - V curves without graphene, which do not show the two properties stated above. Note that the current in Fig. R6d is significantly higher than that in Figs. R6a to c. This is due to the shortened tunneling distance by removal of the graphene layer.

In all these stacking types calculated, the $\sim 1.8\%$ lattice constant mismatch between graphene and BN was not considered following the common treatment in other theoretical works [e.g. ACS Nano, 11, 6382 – 6388 (2017); Adv. Funct. Mater. 33, 2301105 (2023)]. Taking this small mismatch into account would result in a large supercell, which would be very challenging for the first-principles calculations.

Fig. R6. Calculated I - V curves of different stacking configurations. a I - V curves of N-C stacking. **b** I - V curves of B-C stacking. **b** I - V curves of the completely mismatched configuration. **d** I - V curves without the graphene layer.

Changes made: In the second paragraph of the section “**Resistance evolution in different domains**”, the sentence “The stacking energy calculations, as depicted in Fig. S4a, reveal that regardless of trilayer BN states, the boron-on-carbon (B-C) stacking configuration between the bottom graphene and BN is more energetically favorable than the nitrogen-on-carbon (N-C) case, in accordance with a previous study.” has been changed to “**Four stacking types between the trilayer BN and graphene are considered,**

which are boron-on-carbon (B-C) stacking, nitrogen-on-carbon (N-C) stacking, fully aligned and completely mismatched (Figs. S4a to d). The stacking energy calculations, as depicted in Fig. S4e, reveal that regardless of trilayer BN states the B-C stacking is more energetically favorable than others, in accordance with previous studies [ACS Nano 11, 6382-6388 (2017), Adv. Funct. Mater. 33, 2301105 (2023)]. As a result, B-C stacking is adopted to calculate the I - V curves. ”.

The new Fig. S4 is displayed below for your convenience.

Fig. S4 Different stacking configurations and their corresponding energies. a-d Top and side views of the four stacking types at the BN and graphite interface considered in this work, which are B-C stacking, N-C stacking, fully aligned (FA) and competently mismatched (CM), respectively. e Calculated stacking energies of these four types for the four different polarization combinations. B-C stacking always has the lowest energy.

The following discussion about other stacking types is added after this paragraph.

“Since the BN and graphite are not intentionally aligned in the experiment, other stacking types could also present in the devices. Therefore, we calculated I - V curves for the N-C stacking (Fig. S6a), fully aligned (Fig. S6b) and completely mismatched (Fig.S6c) types. All of them, similar to the most stable B-C stacking configuration, show the same behaviors and agree qualitatively with the experiment.”

The new Fig. S6 is displayed below for your convenience.

Fig. S6. Calculated I - V curves of different stacking configurations. a I - V curves of N-C stacking. **b** I - V curves of B-C stacking. **b** I - V curves of the completely mismatched configuration. **d** I - V curves without the graphene layer.

Comment 2-4. The authors performed theoretical calculations to explain the reason why graphite-BN interface dominates the signal contrast. Although the theoretical calculations seem to agree with the experiment, I wonder if the authors can also give some intuitive arguments in the manuscript. Is it mainly because the tip can be considered as a perfect metal, while the band structure in graphite affects the tunneling probability depending on the polarization of nearest BN bilayer?

Response 2-4: We thank the reviewer for this insightful comment. We can analyse the charge transfer process at the BN-graphene interface to gain more understanding.

Due to the difference in electronegativity, there exists charge transfer between BN and graphene. At a first glance, the impact of the graphene substrate on the trilayer BN should be identical for all four polarization states since the stacking between the bottom BN and graphene is fixed (e.g. as the energetically favored B-C configuration) in one calculation. The situation is changed, however, if the across-layer effect is taken into account [experiment: Nature 588, 71-76 (2020); theory: Adv. Funct. Mater. 33, 2301105 (2023); Phys. Rev. Lett. 131, 096801 (2023)]. The lower two layers of BN together determines the charge transfer process.

The $\uparrow\uparrow$ and $\downarrow\uparrow$ polarization states share the same \uparrow polarization formed by the lower two BN layers. As a result, their electronic environments at the BN/graphene interface are similar yet very different from those of the $\downarrow\downarrow$ and $\uparrow\downarrow$ states. As displayed in Fig. R7, the N atoms of the $\uparrow\uparrow$ and $\downarrow\uparrow$ states lose electrons (blue isosurfaces) on their sides facing graphene. In contrast, the N atoms of the $\downarrow\downarrow$ and $\uparrow\downarrow$ group accumulate electrons (yellow isosurfaces). We then performed the Hirshfeld charge analysis of the bottom BN layer. For the $\uparrow\uparrow$ and $\downarrow\uparrow$ polarization states, B/N atoms have charges of $\sim 0.205e/-0.204e$, making the BN layer almost electroneutral. On the other hand, the bottom BN layers in the $\downarrow\downarrow$ and $\uparrow\downarrow$ polarization states carry a net charge of $\sim -0.008e$ per unit cell.

Our differential charge density distributions, in accompany with Hirshfeld charge analysis, clearly demonstrate that due to the across-layer effect, the impact of graphene substrate on the trilayer BN differs depending on the polarization state of the lower two layers of BN.

Fig. R7. Differential charge density distributions where yellow and blue isosurfaces indicate electron accumulation and depletion after layer stacking, respectively. The Hirshfeld charges for the BN layer neighboring to graphene are labeled.

Changes made: For the second paragraph of “**Resistance evolution in different domains**”, the sentence “It is clear that the polarization residing between the lower two BN layers determines this overall behavior, corroborating experimental observation.” has been expanded as follows.

“It is clear that the polarization residing between the lower two BN layers determines this overall behavior, which can be seen more clearly in the charge distribution at the BN/graphene interface. With the across-layer effect [Adv. Funct. Mater. 33, 2301105 (2023); Phys. Rev. Lett. 131, 096801 (2023)] taken into account, charge distribution depends on the interaction between the bottom two layers of BN and graphene substrate. As shown in Fig.S5, differential charge density distributions, in accompany with Hirshfeld charge analysis, confirm that the impact of the graphite substrate on trilayer BN differs depending on the polarization state of the lower two layers of BN, corroborating experimental observations.”

The newly added charge distribution is inserted into the Supplementary Information as Fig. S5 and discussed as follows.

Fig. S5. Differential charge density distributions where yellow and blue isosurfaces indicate electron accumulation and depletion after layer stacking, respectively. The Hirshfeld charges for the BN layer neighboring to graphene are labeled.

The $\uparrow\uparrow$ and $\downarrow\uparrow$ polarization states share the same \uparrow polarization formed by the lower two BN layers. As a result, their electronic environments at the BN/graphene interface are similar yet very different from those of the $\downarrow\downarrow$ and $\uparrow\downarrow$ states. As displayed in Fig. S5, the N atoms of the $\uparrow\uparrow$ and $\downarrow\uparrow$ states lose electrons (blue isosurfaces) on their sides facing graphene. In contrast, the N atoms the $\downarrow\downarrow$ and $\uparrow\downarrow$ group accumulate electrons (yellow isosurfaces). We then performed the Hirshfeld charge analysis of the bottom BN layer. For the $\uparrow\uparrow$ and $\downarrow\uparrow$ polarization states, B/N atoms have similar charges of $\sim 0.205e/-0.204e$, making the BN layer almost electroneutral. On the other hand, the bottom BN layer in the $\downarrow\downarrow$ and $\uparrow\downarrow$ polarization states carries charge of $\sim -0.008e$ per unit cell.

Our differential charge density distributions, in accompany with Hirshfeld charge analysis, clearly demonstrate that due to the across-layer effect, the impact of graphene substrate on the trilayer BN differs depending on the polarization state of the lower two layers of BN.

Reviewer #3 (Remarks to the Author):

The authors show that different polarisation domains can exist in three rotated hBN layers.

The work is interesting, although the results could be better explained.

Response: We express our sincere appreciation to the reviewer for his/her positive feedback on our manuscript.

I have some questions or comments for the authors.

Comment 3-1. The stacking between the three hBN layers is not completely clear; apparently, between the 1-2 is 0.3° , but the angle between the 2-3 is not indicated.

Response 3-1: We thank the reviewer for pointing out this confusing part. The target twist angles between 1-2 and 2-3 layers are both 0.3° . However, due to wrinkles and bubbles in the fabricated samples, the twist angle varies slightly. We calculated the angles in the samples studied from the Moire pattern period to be $0.03^\circ \sim 0.24^\circ$.

Changes made: We have added the following discussion to the Methods section.

“Due to the wrinkles formed in the stacking process, the final twist angles between the first and second (second and third) layers of BN varied among different areas in the samples, and were determined from the Moire pattern period to be $0.03^\circ \sim 0.24^\circ$. ”

Comment 3-2. The notation of the regions with up/down arrows can be confusing as the right arrow would indicate the interface; the notation leads intuitively to think of areas with aligned or unaligned polarisations. For instance, I would not know what symbology to use when polarisations are opposite at different interfaces, creating a net zero polarization.

Comment 3-3. Labeling the stacks in different regions would be suitable according to the relative stacking between the hBN layers. For example, the ABA would not produce any net polarisation by symmetry, but an ABC or ACB would do.

Comment 3-4. The above can also be applied to DFT calculations (Fig. 4).

Response to Comments 3-2, 3-3 and 3-4: We thank the reviewer for these three very important comments regarding the notations, which help us to improve the clarity of our data presentation. We have added a table in the Supplementary Information to clarify the notation used in the manuscript, which is shown below for your convenience. The table is mentioned in Page 2 of the manuscript where the notations first appear.

ABC	BAB	CBA	ABA
			↑↑	↓↑	↓↓	↑↓

Table. S1 Polarization notation definitions. Each column represents three equivalent notations for a particular stacking and polarization state.

Comment 3-5. In the calculations of Fig S4, are the same lattice vectors used for hBN and graphene? The most probable scenario is that the two crystals are not aligned.

Response 3-5. We thank the reviewer for this very important comment. Indeed the same lattice vectors were used in the calculations. It has been experimentally observed that for small-angle twisted graphene/hBN heterostructures, lattice reconstruction occurs to form commensurate state [Nat. Phys. 10, 451-456 (2014)], where strain is accumulated in domain walls that separate graphene/hBN areas with matching lattice constants. However, just as the reviewer pointed out, since in our experiment the BN and graphite were not intentionally aligned, the most probable scenario is that the two crystals are not commensurate. Nevertheless, in first-principles calculations of the graphene/hBN system, it is customary to ignore the ~1.8% difference and use the same lattice constant for both graphene and hBN [e.g. ACS Nano, 11, 6382 - 6388 (2017); Adv. Funct. Mater. 33, 2301105 (2023)]. If the ~1.8% difference was taken into account, large supercells would be needed to construct the lattice model which post great challenges for the first-principles calculations. As a result, we follow the literature and take the same lattice constant in our calculation. Even though the lattice models we used may be

approximative, the theoretical results still provide invaluable insights into our experiment, i.e. without graphene the I - V curves would not form two groups and no crossing would be expected.

To better address this comment and related ones from other reviewers, we calculated I - V curves for other three typical stacking orders, i.e. N-C stacking, fully aligned, and completely mismatched. The corresponding lattice configurations are shown below in Fig. R8.

Fig. R8. Top and side views of the atomic arrangements between the bottom BN and the graphene layers, showing the B-C stacking, N-C stacking, fully aligned and completely mismatched configurations.

The calculated I - V curves are plotted below as Fig. R9. It can be seen from Figs. R9a to c that all these stacking orders give similar I - V behaviors although details vary between them. First, the four curves form two groups, and the polarization from the lower two BN layers plays a dominant role in affecting the current. Second, within each group the I - V curves can cross, so that the contrast in current mappings can change as the sample bias voltage is swept. These results agree qualitatively with experimental observations. As a comparison, Fig. R9d plots the I - V curves without graphene, which do not show the two properties stated above. Note that the current in Fig. R9d is significantly higher than that in Figs. R9a to c. This is due to the shortened tunneling distance by removal of the graphene layer.

Fig. R9. Calculated I - V curves of different stacking configurations. a I - V curves of N-C stacking. **b** I - V curves of B-C stacking. **b** I - V curves of the completely mismatched configuration. **d** I - V curves without the graphene layer.

Changes made: In the second paragraph of the section “**Resistance evolution in different domains**”, the sentence “The stacking energy calculations, as depicted in Fig. S4a, reveal that regardless of trilayer BN states, the boron-on-carbon (B-C) stacking configuration between the bottom graphene and BN is more energetically favorable than the nitrogen-on-carbon (N-C) case, in accordance with a previous study.” has been changed to “**Four stacking types between the trilayer BN and graphene are considered,**

which are boron-on-carbon (B-C) stacking, nitrogen-on-carbon (N-C) stacking, fully aligned and completely mismatched (Figs. S4a to d). The stacking energy calculations, as depicted in Fig. S4e, reveal that regardless of trilayer BN states the B-C stacking is more energetically favorable than others, in accordance with previous studies [ACS Nano 11, 6382-6388 (2017), Adv. Funct. Mater. 33, 2301105 (2023)]. As a result, B-C stacking is adopted to calculate the $I-V$ curves. ”.

The new Fig. S4 is displayed below for your convenience.

Fig. S4 Different stacking configurations and their corresponding energies. a-d Top and side views of the four stacking types at the BN and graphite interface considered in this work, which are B-C stacking, N-C stacking, fully aligned (FA) and competently mismatched (CM), respectively. **e** Calculated stacking energies of these four types for the four different polarization combinations. B-C stacking always has the lowest energy.

The following discussion about other stacking types is added after this paragraph.

“Since the BN and graphite are not intentionally aligned in the experiment, other stacking types could also present in the devices. Therefore, we calculated $I-V$ curves for the N-C stacking (Fig. S6a), fully aligned (Fig. S6b) and completely mismatched (Fig.S6c) types. All of them, similar to the most stable B-C stacking configuration, show the same behaviors and agree qualitatively with the experiment.”

The new Fig. S6 is displayed below for your convenience.

Fig. S6 Calculated I - V curves of different stacking configurations. a I - V curves of N-C stacking. **b** I - V curves of B-C stacking. **b** I - V curves of the completely mismatched configuration. **d** I - V curves without the graphene layer.

Comment 3-6. Can you explain why, in the hBN bilayer region, there is no dependence of the tunneling current on the lower graphene layer?

Response 3-6. We thank the reviewer for giving us the opportunity to clarify this point. In the hBN bilayer region, the tunneling current also sensitively depends on the lower graphene layer. To prove this, we replace the graphene with another material, a semiconducting MoSe₂. Fig. R10a shows the I - V curves of a graphene substrate sample,

while Fig. R10b shows the same measurements of a MoSe₂ substrate sample. Drastic differences can be seen in these two samples. First, the resistance order is switched when the substrate is changed. In Fig. R10a, the low current state corresponds to polarization pointing down, while in Fig. R10b, the low current state corresponds to polarization pointing up. Second, there is a gap like feature in Fig. R10b, which is absent in Fig. R10a. If a larger bias voltage is applied to induce polarization switching as shown in Figs. R10c and d, the hysteresis windows also behave very differently. For the graphene substrate sample (Fig. R10c), the current jump from a low current state to a high current state. For the MoSe₂ substrate sample (Fig. R10d), however, the switching path is opposite. These comparisons clearly show that the substrate has a significant impact on the tunneling behavior and worth further study.

[Redacted]

Fig. R10. Comparison between twosited-bilayer BN samples with graphite and MoSe₂ substrate. a, b I - V curves on domains of different polarization with graphite and MoSe₂ substrates, respectively. **c, d** Polarization switching on samples with graphite and MoSe₂ substrates, respectively.

Comment 3-7. There is an error in the labeling of Fig S6.

Response 3-7. We sincerely thank the reviewer for the very careful reading. The labeling has been corrected.

REVIEWER COMMENTS

Reviewer #1 (Remarks to the Author):

I recommend publishing the work as it effectively addresses all issues with reasonable explanations and offers valuable insights into the construction of multiresistance state device systems. There is a minor spelling error to note 'moiré pattern'.

Reviewer #2 (Remarks to the Author):

Please see the attachment.

Reviewer #3 (Remarks to the Author):

The authors have answered the questions/comments.
I believe it can be published.

The authors have answered my questions in detail and improved their manuscript. However, I still have a few concerns that prevent me from recommending publication of the paper.

1. The authors have refined their references, but there are still some missing. For example, “Kim, D.S., Dominguez, R.C., Mayorga-Luna, R. *et al.* Electrostatic moiré potential from twisted hexagonal boron nitride layers. *Nat. Mater.* (2023)” should be included. Please do a more thorough literature review.
2. What is the main novelty of this work? Is it the manipulation of anti-ferroelectric states? Given these previous literatures that observe the anti-ferroelectric states, I think it is important for the authors to give credit to the previous observations and then emphasize the novelty of this work. So the audience will be able to capture the main new findings of the paper.
3. The authors gave a better explanation of identifying the polarized states. However, the following figure is still quite confusing to me. For example, is the red arrow denoting the green circled one or red circled one? Is the boundary in blue line that helped to distinguish the up or down states on the upper layer? This image has complicated structures and a high resolution and clear denotation is crucial for understanding.

4. In Fig. S2a, b, the defect moved a lot. Is it because the defect moved, or there was a significant thermal drift? How do I know that the triangular region shrank if there was such a large thermal drift?

Response letter to reviewers
(Article No. NCOMMS-23-44160A)

We want to express our deepest gratitude to all three reviewers for their support of our manuscript. We have revised the manuscript to address the remaining issues raised by the reviewers.

Reviewer #1 (Remarks to the Author):

I recommend publishing the work as it effectively addresses all issues with reasonable explanations and offers valuable insights into the construction of multiresistance state device systems. There is a minor spelling error to note ‘moiré pattern’.

Response: We thank the reviewer for the recommendation of publication of our work! We have correct this typo in the Methods section.

Reviewer #2 (Remarks to the Author)

The authors have answered my questions in detail and improved their manuscript. However, I still have a few concerns that prevent me from recommending publication of the paper.

Comment 2-1. The authors have refined their references, but there are still some missing. For example, “Kim, D.S., Dominguez, R.C., Mayorga-Luna, R. et al. Electrostatic moiré potential from twisted hexagonal boron nitride layers. *Nat. Mater.* (2023)” should be included. Please do a more thorough literature review.

Response 2-1: We thank the reviewer for bringing this important literature to our attention. It is indeed crucial to keep track of all the important progresses in a rapidly developing field such as the interfacial ferroelectricity.

We have searched through the ~360 papers that cited the pioneer experimental works demonstrating ferroelectricity in twisted boron nitride [*Science* 372, 1458 (2021), *Science* 372, 1462 (2021), *Nat. Commun.* 12, 347 (2021)], and added the following papers about trilayer interfacial ferroelectricity/antiferroelectricity to the manuscript. The newly added references, together with other related ones already cited in the previous submission, are discussed in the revised manuscript to highlight the novelty of this work.

14. Kim, D.S., Dominguez, R.C., Mayorga-Luna, R. et al. Electrostatic moiré potential from twisted hexagonal boron nitride layers. *Nat. Mater.* (2023)

15. Wu, C.L., Wong, S.S., Lin, Z.Y. et al. Epitaxially ferroelectric hexagonal boron nitride on graphene. Preprint on *Research Square*. DOI: <https://doi.org/10.21203/rs.3.rs-2686336/v1>

Comment 2-2. What is the main novelty of this work? Is it the manipulation of anti-ferroelectric states? Given these previous literatures that observe the anti-ferroelectric states, I think it is important for the authors to give credit to the previous observations and then emphasize the novelty of this work. So the audience will be able to capture the main new findings of the paper.

Response 2-2: We thank the reviewer for this very important suggestion which helps us to further improve the manuscript. Just as the reviewer pointed out, antiferroelectric states in the interfacial ferroelectric systems have been realized in several papers published very recently, including one work on twisted trilayer WSe₂ and MoS₂ [*Nature* 612, 465 (2022)], and two works on twisted trilayer hBN [*Nat. Mater.* (2023) and the preprint on *Research Square* mentioned in Response 2-1]. Related theoretical papers also discussed the mechanism of the multilayer ferro-/antiferroelectric states [*J. Phys. Chem. Solids* 173, 111086 (2023) and *Phys. Rev. Lett.* 131, 096801 (2023)]. Compared with the published experimental realizations of the interfacial antiferroelectric systems, the novelty of the current work can be summarized as follows.

1. In the previous experimental works, they showed the existence of both the ferro- and antiferroelectric states in twisted trilayer systems with surface potential measurements [*Nature* 612, 465 (2022), *Nat. Mater.* (2023)], and the antiferroelectric state in the epitaxially grown trilayer hBN with piezoelectric force microscopy [the preprint on *Research Square*]. Our work is the first one using the trilayer hBN as a tunneling junction, which can probe the rich polarization states and switching mechanism unique to this system, and reveal the application potentials such as in memories based on ferroelectric tunneling junctions.

2. In this work, we showed that the polarization states in the interfacial ferro- and antiferroelectric BN are switched in a layer-by-layer process, which is rare in other ferro- and antiferroelectric systems. We also precisely identified all the possible switching pathways and discovered the interesting, albeit weak, interaction between the polarization states located at the different interfaces.

3. The multi-resistant states resulted from the layer-by-layer switching process can be utilized in novel devices such as multi-valued memories, as noted by Reviewer #1 in the last reviewing process, “This work may be helpful to develop an effective approach for building multiresistance states system in the very interesting 2D samples.”

To help the readers better capture the main new findings of this work, we have added the following discussions about the closely related previous works and the novelty of the current work in the introduction paragraph of the revised manuscript.

Changes made: The newly added discussions are shown in red.

In addition, the layer-stacking method is very flexible as compared with traditional crystal growth. Therefore, it is foreseeable that by stacking more layers different polarization states could be obtained, which have been observed in multilayer 3R MoS₂ samples [*Nat. Commun.* 13, 7696 (2022), *Nano Lett.* 23, 7228-7235 (2023)], manually stacked or epitaxially grown trilayer BN samples [*Nat. Mater.* (2023), preprint on *Research Square*], manually stacked trilayer WSe₂ and MoS₂ [*Nature* 612, 465 (2022)]. Here, we fabricate trilayer BN samples and unveil the rich polarization states together with their intriguing switching behaviors in this system with the ferroelectric tunneling process. The ferro- and antiferroelectric domains developed in the trilayer BN samples exhibit layer-by-layer switching behavior, which is rare in other ferro- and antiferroelectric systems. All the possible switching pathways are precisely identified, which reveals the interesting, albeit weak, interaction between the polarization states located at the different interfaces. The multiresistance states resulted from the layer-by-layer switching process can be utilized in novel devices such as multi-valued memories. With first-principles calculations, we find that not only the internal polarization of the trilayer BN but also the graphite substrate are important in explaining the observed resistance states.

Comment 2-3. The authors gave a better explanation of identifying the polarized states. However, the following figure is still quite confusing to me. For example, is the red arrow denoting the green circled one or red circled one? Is the boundary in blue line that helped to distinguish the up or down states on the upper layer? This image has complicated structures and a high resolution and clear denotation is crucial for understanding.

Response 2-3: We apologize for the confusion caused by this figure. We have replotted the figures below as Fig. R1. The blue dashed lines outline the domains from the bottom two layers of BN, and the red dashed lines outline one domain from the top two layers of BN. The blue arrows denote the bottom polarization states, and the red arrows denote the top ones. In the trilayer area, each domain is marked with one red and one blue arrows to show the polarization combinations.

Fig. R1. CAFM mapping near a step between the bilayer-trilayer BN. a and b CAFM images around the step at 100 mV sample bias voltage. These two images are from the same scan but displayed with different color scales, since the bilayer and trilayer areas have an order of magnitude difference in current. The blue dashed lines outline the domains from the bottom two layers of BN, and the red dashed lines outline one domain from the top two layers of BN.

Changes made: Figure S1 in the Supplementary Information is replaced by the figure below and the newly added description is shown in red.

Fig. S1 CAFM mapping near a step between the bilayer and trilayer BN. **a** and **b** CAFM images around the step at 100 mV sample bias voltage. These two images are from the same scan but displayed with different color scales, since the bilayer and trilayer areas have an order of magnitude difference in current. **The blue dashed lines outline the domains from the bottom two layers of BN, and the red dashed lines outline one domain from the top two layers of BN.** **c** Corresponding KPFM image around the step with a larger scanning area. The white dashed box corresponds to the area shown in **a** and **b**. **d** Profiles of the potential along the green arrow in **c**.

Comment 2-4. In Fig. S2a, b, the defect moved a lot. Is it because the defect moved, or there was a significant thermal drift? How do I know that the triangular region shrank if there was such a large thermal drift?

Response 2-4: We thank the reviewer for this important observation. In Figs. S2a, b (which are shown below for your convenience), the defect appeared at different locations due to drift in the scan. Drift, caused by creep or hysteresis of the scanner and thermal drift or vibration of the instrument, is a common artifact in scanning probe microscopy techniques [e.g. *Measurement*, 159, 107776, (2020)]. In the instrument we used (Asylum Cypher AFM), the drift in ~10 min scan is ~10 nm, which becomes significant when the scanning area is comparable to the drift. In Figs. S2a and b, a small scan area is shown to better demonstrate the contrast change. The triangular domain in the upper left corner can be seen to have shrunk from Fig. S2a to Fig. S2b with an observable size reduction. The domain in the lower right corner appears to have the same

size in Figs. Sa and b. However, with the help of the defect (orange circle), we can see that the horizontal domain wall (blue dashed line) has actually moved up.

Fig. S2 An example of using the SKPM to identify polarization states. **a** and **b** CAFM images at 100 mV and 500 mV sample bias. The orange circles in **a** and **b** highlight a defect to show the thermal drift during scan. The ferroelectric domains can be easily identified from the change of their shapes, and are marked out in **b**. The triangular domain in the upper left corner has an observable size reduction, while the domain in the lower right corner appears to be the same from **a** to **b**. This is due to the drift effect. With the defect as a reference, we can see that its horizontal domain wall has moved upwards.

In this study, we usually start from a larger scan area to locate the region of interest, where the change of domain sizes is more obvious. Fig. 2 of the main text (which is shown below for your convenience) shows one such example. Due to the larger scan area, drift (~8 nm) is less apparent and the domain wall evolution is easier to identify.

Fig. 2 Evolution of the domains under an external electric field. **a** and **b** CAFM images with 100 mV and 500 mV sample bias voltages. Two small triangular domains are highlighted by the red dash lines. The blue dashed line marks a domain wall from a different interface. The orange circle points out a surface defect that can be used as a reference of position since the instrument has unavoidable drift during scan.

Changes made: In Fig. S2, we have revised the figure caption (shown in red) to better discuss the drift effect.

Fig. S2 An example of using the SKPM to identify polarization states. a and b CAFM images at 100 mV and 500 mV sample bias. The orange circles in **a** and **b** highlight a defect to show the thermal drift during scan. The ferroelectric domains can be easily identified from the change of their shapes, and are marked out in **b**. **The triangular domain in the upper left corner has an observable size reduction, while the domain in the lower right corner appears to be the same from a to b. This is due to the drift effect. With the defect as a reference, we can see that its horizontal domain wall has moved upwards.**

Reviewer #3 (Remarks to the Author):

The authors have answered the questions/comments.

I believe it can be published.

Response: We thank the reviewer for the recommendation of publication of our work!

REVIEWERS' COMMENTS

Reviewer #2 (Remarks to the Author):

The authors have nicely addressed all my questions, and I would like to recommend publication of this interesting work in Nature Communications.